# Simultaneous Analysis of Free/Combined Phytosterols in Rapeseed and Their Dynamic Changes during Microwave Pretreatment and Oil Processing

**DOI:** 10.3390/foods11203219

**Published:** 2022-10-14

**Authors:** Dong Li, Dan Wang, Huaming Xiao, Xin Lv, Chang Zheng, Changsheng Liu, Hong Chen, Fang Wei

**Affiliations:** Oil Crops Research Institute of Chinese Academy of Agricultural Sciences, Key Laboratory of Oilseeds Processing of Ministry of Agriculture, Key Laboratory of Biology and Genetic Improvement of Oil Crops of Ministry of Agriculture, Hubei Key Laboratory of Lipid Chemistry and Nutrition, Wuhan 430062, China

**Keywords:** phytosterol, rapeseed, microwave pretreatment, oil processing

## Abstract

Here, a simple, efficient, and rapid solid phase extraction-gas chromatography (SPE–GC) method was developed for the simultaneous analysis of free/combined phytosterols in rapeseed and their dynamic changes during microwave pretreatment and oil processing. First, by comparing different methods for extracting free/combined phytosterols from rapeseed and rapeseed cake, the Folch method was considered to be the optimal method and was selected in subsequent experiments. Subsequently, the extraction method was validated by determining the recoveries of standards (brassinosterol, campesterol, β-sitosterol and cholesteryl oleate) spiked in rapeseed and rapeseed oil samples, and the recoveries were in the range from 82.7% to 104.5% and 83.8% to 116.3%, respectively. The established method was applied to study the dynamic changes of the form and content of phytosterols in rapeseed and its products (rapeseed oil and cake) during rapeseed microwave pretreatment and the oil production process. Additionally, the results showed that more than 55% of the free/combined phytosterols in rapeseed were transferred to rapeseed oil during the oil processing, and this proportion will increase after microwave pretreatment of rapeseed. This work will provide analytical methods and data support for a comprehensive understanding of phytosterols in rapeseed and its products during oil processing.

## 1. Introduction

As among the four major oil crops in the world, rapeseed is the main source of edible vegetable oil and vegetable protein [1,2]. Rapeseed is rich in lipids, which can provide the body with energy, essential fatty acids and various lipid concomitants, such as phytosterols, vitamin E, and squalene. [3]. Rapeseed oil and rapeseed cake are produced through rapeseed processing, in which rapeseed oil is an important source of health-related compounds such as polyphenols, phytosterols, and tocopherols in the human diet [4], while rapeseed cake provides the base material for a variety of commercial products, such as feed and fertilizer [5,6,7].

Phytosterols are a class of tetracyclic triterpenoids that regulate the physicochemical functions of plant cell membranes, and their chemical structures are similar to cholesterol [8]. The difference is that phytosterols are unable to be synthesized endogenously in the human body and can only be obtained from the diet through intestinal absorption [8]. Vegetable oils are among the major dietary sources of phytosterols [4]. Free and combined phytosterols are common forms of phytosterols in plants, wherein combined phytosterols include phytosterol fatty acid esters, phytosterol phenolic acid esters, phytosterol glycosides and acylated phytosterol glycosides (Figure 1) [9].

Phytosterols can effectively manage insulin metabolism and cholesterol regulation processes, and can reduce inflammation and oxidative stress damage associated with DNA repair mechanisms [4]. Therefore, phytosterols (excluding brassicasterol) have a positive effect on the prevention of cardiovascular disease and diabetes, and the FDA (Food and Drug Administration) has approved a health claim for phytosterols as cholesterol-lowering agents [10,11,12,13]. The biological function of phytosterols mainly depends on free phytosterols [14].

Free phytosterols in vegetable oils are more easily oxidized than combined phytosterols, and oxidized phytosterols have potential negative effects on human health [15,16,17,18]. In addition, due to the low oil solubility of free phytosterols, their practical effects on human health are hindered [19,20]. Phytosterols in the form of esters are expected as alternatives because phytosterol esters have higher oil solubility [21]. Moreover, phytosterol esters can be decomposed into fatty acids and free phytosterols through the action of pancreatic cholesterol lipase in the human intestinal tract, to play the role of free phytosterols and still maintain the ability to lower cholesterol [22,23,24,25,26].

The analytical methods of phytosterols include immunoassay, ferrophosphorus reagent method, spectroscopy, chromatography, and mass spectrometry [27,28,29,30,31]. Chen et al. developed a gas chromatography-mass spectrometry (GC-MS) method to study the content and composition of total phytosterols, free phytosterols and esterified phytosterols in lotus seed oil, and the results showed that the total phytosterols of lotus plumule oil were 12.10–14.21 g/100 g [32]. Phillips A et al. successfully determined the concentration of free phytosterols and esterified phytosterols (including campesterol, β-sitosterol, stigmasterol, Δ^5^-avenasterol, brassicasterol, sitostanol, campestanol and cholesterol) in 31 edible oils and fats using solid phase extraction-gas chromatography (SPE–GC) method [33].

As an efficient pretreatment method, microwave technology has a very important application in rapeseed processing [34]. The contents of polyphenols, phytosterols and vitamin E in rapeseed oil can be significantly increased by microwave pretreatment before rapeseed pressing [35,36]. In addition, microwave pretreatment can improve the oil yield of rapeseed and accelerate the degradation of glucosinolates, making rapeseed oil richer in aroma [37,38,39,40].

It is of great significance to study the dynamic changes of the form and content of phytosterols in rapeseed and its products (rapeseed oil and cake) during rapeseed microwave pretreatment and the oil production process for rational processing and dietary rapeseed oil. However, there are still few studies on the changes of different forms of phytosterols during rapeseed oil processing. In this study, a SPE–GC method was used to simultaneously analyze the contents of free and combined phytosterols in rapeseed, rapeseed oil and rapeseed cake. The dynamic changes of the form and content of phytosterols in rapeseed and its products (rapeseed oil and cake) during rapeseed microwave pretreatment and the oil production process were also investigated.

## 2. Materials and Methods

### 2.1. Materials and Reagents

Two varieties of rapeseeds, Zhongyouza 19 and Dadi 199, provided by Oil Crops Research Institute, Chinese Academy of Agricultural Sciences; pressed rapeseed oil (first-grade, Arowana); *n*-hexane (chromatographic grade, Fisher company, London, UK); ethyl acetate, chloroform, ethanol, potassium hydroxide, anhydrous sodium sulfate, H_2_SO_4_ and NaCl (analytical grade, Sinopharm Chemical Reagent Co., Ltd., Shanghai, China); *N*,*O*-bis(trimethylsilyl)trifluoroacetamide (containing 1% trimethylchlorosilane, TCI); β-cholestanol (CATO); brassicasterol, campesterol, β-sitosterol, cholesterol oleate and methyl heptadecanoate standard (Shanghai Anpu Cui Shi Standard Technology Service Co., Ltd., Shanghai, China).

### 2.2. Instruments

A 6890 gas chromatograph system with flame ionization detector (FID) (Agilent Technologies, Santa Clara, CA, USA); LTP-205 oil press (Liangtai, Dongguan Xiangju Intelligent Co., Ltd., Dongguan, China); MTV-100 multi-tube vortex mixer (Hangzhou Aosheng Instrument Co., Ltd., Hangzhou, China); DC-24 nitrogen blow dryer (Shanghai Anpu Experiment Technology Co., Ltd., Shanghai, China); closed microwave digestion apparatus (CEM, St. Louis, MO, USA); constant temperature water bath magnetic stirrer (HH-6D, Changzhou Zhongcheng Instrument Manufacturing Co., Ltd., Changzhou, China); electronic analytical balance (ML204, Mettler Toledo Instruments (Shanghai) Co., Ltd., Shanghai, China); refrigerated centrifuge (Thermo Fisher Scientific, Waltham, MA, USA); electric heating constant temperature blast drying oven (SHFG-01); silica SPE cartridge with 500 mg sorbent per cartridge (Sep-Pak, Waters, Milford, CT, USA); YTLG-10A vacuum freeze dryer (Shanghai Yetuo Technology Co., Ltd., Shanghai, China).

### 2.3. Methods and Procedures

Schematic diagram for analysis the content of free and combined phytosterols in rapeseed and its products (rapeseed oil and cake) during rapeseed microwave pretreatment and the oil production process is shown in Figure 2.

#### 2.3.1. Sample Pretreatment


(1)Two varieties of rapeseed, Zhongyouza 19 and Dadi 199, were pretreated with microwave, referring to the method of Q. Zhou [39] and others [34,38] with slight modifications. After cleaning and impurity removal, the rapeseed was adjusted to 10% moisture, placed in refrigerator at 4 °C for 12 h, weighed 400 g, divided into 8 plates with a diameter of 9 cm, and placed in a microwave oven at a microwave power of 800 W for 7 min, quickly cooled to room temperature, and rapeseed without microwave treatment was used as a blank control.(2)Drying of rapeseed. An appropriate amount of rapeseed was pre-frozen in an ultralow temperature refrigerator for 6 h, and then freeze-dried by a vacuum freeze dryer to obtain a dry base of rapeseed, which was stored in refrigerator at −20 °C until further process.(3)Preparation of cold-pressed canola oil (two varieties of rapeseeds, Zhongyouza 19 and Dadi 199). Dried non-microwave rapeseed and microwaved rapeseed were placed in refrigerator at 4 °C for 12 h, and the moisture content was adjusted to 6 %. Take 100 g dried non-microwave rapeseed and microwaved rapeseed and press with LTP-205 Liangtai oil press to obtain rapeseed oil and rapeseed cake, which are stored in refrigerator at 4 °C until further process.


#### 2.3.2. Extraction of Phytosterols from Rapeseed and Rapeseed Cake

An appropriate amount of rapeseed or rapeseed cake was pre-frozen in refrigerator at −80 °C for 6 h, and then freeze-dried by a vacuum freeze dryer to obtain a dry rapeseed, which was stored in refrigerator at −20 °C. An appropriate amount of freeze-dried rapeseed or rapeseed cake was crushed into fine powder by a universal pulverizer, transferred to a 30 mL glass tube and placed in refrigerator at −20 °C until further process. Three liquid–liquid extraction methods (including Folch, Bligh–Dyer and methyl tert-butyl ether (MTBE)) were employed to extract phytosterols from rapeseed and rapeseed cake, respectively, and the extraction efficiencies were compared. The phytosterols obtained were stored at −20 °C until further analysis.

(1) The Folch extraction method was carried out according to a modified procedure of Folch, Lees, and Stanley [41]. Briefly, 0.2 g crushed powder (rapeseed or rapeseed cake) was completely mixed with 2.2 mL methanol, and then 4.4 mL chloroform were added. Then, β-cholestanol (100 μL, 1 mg/mL in *n*-hexane) was added as an internal standard. The mixture was vortexed at 1800 rpm for 5 min, sonicated at 30 °C for 1 h, and then 1.65 mL deionized water was added, vortexed at 1800 rpm for 10 min, centrifuged at 5000 rpm for 5 min, and then the organic phase was collected. Add chloroform to the aqueous phase and extract twice (2 × 2 mL), and then the collected organic phases were combined, and dried at 35 °C by using a gentle stream of nitrogen to recover the phytosterols. Finally, 5 mL *n*-hexane was added to reconstitute and stored for further use.

(2) The Bligh–Dyer extraction method was carried out according to a modified procedure of Bligh and Dyer [42]. Briefly, 0.2 g crushed powder (rapeseed or rapeseed cake) was completely mixed with 3.25 mL methanol, and then 3.25 mL chloroform were added. Then, 100 μL β-cholestanol (1 mg/mL in *n*-hexane) was added as an internal standard. The mixture was vortexed at 1800 rpm for 5 min, sonicated at 30 °C for 1 h, and then 2 mL deionized water was added, vortexed at 1800 rpm for 10 min, centrifuged at 5000 rpm for 5 min, and then the organic phase was collected. Add chloroform to the aqueous phase and extract twice (2 × 2 mL), and then the collected organic phases were combined, and dried at 35 °C by using a gentle stream of nitrogen to recover the phytosterols. Finally, 5 mL *n*-hexane was added to reconstitute and stored for further use.

(3) The MTBE extraction method was carried out according to a modified procedure of Matyash et al. [43]. Briefly, 0.2 g crushed powder (rapeseed or rapeseed cake) was completely mixed with 5.5 mL of a mixture of MTBE/methanol (2:1, *v*/*v*). Then, 100 μL β-cholestanol (1 mg/mL in *n*-hexane) was added as an internal standard. The mixture was vortexed at 1800 rpm for 5 min, sonicated at 30 °C for 1 h, and then 1.25 mL deionized water was added, vortexed at 1800 rpm for 10 min, centrifuged at 5000 rpm for 5 min, and then the organic phase was collected. Add MTBE to the aqueous phase and extract twice (2 × 2 mL), and then the collected organic phases were combined, and dried at 35 °C by using a gentle stream of nitrogen to recover the phytosterols. Finally, 5 mL *n*-hexane was added to reconstitute and stored for further use.

#### 2.3.3. Separation of Combined and Free Phytosterols by SPE

Rapeseed oil sample can be used directly after dilution: accurately weigh 50 mg of oil sample into a glass test tube, 100 μL β-cholestanol (1 mg/mL in *n*-hexane) was added as an internal standard, finally, 5 mL *n*-hexane was added to dilute the oil sample and stored for further use. 

#### 2.3.4. Separation of Free/Combined Phytosterols by SPE

SPE was carried out referring to the method of Rebecca Esche et al. [44] with a slight modification. 500 mg sorbent per cartridge (Sep-Pak, Waters, St. Louis, MO, USA) was used for SPE, firstly, about 1 g of anhydrous sodium sulfate was added to the top of the SPE column, then activated with 2 × 5 mL of *n*-hexane solution, and the effluent was discarded; then 5 mL of the sample solution obtained in Section 2.3.2 was injected into the SPE column, and the effluent was discarded; after that the combined phytosterols was eluted with 2 × 5 mL of *n*-hexane/ethyl acetate (96:4, *v*/*v*) (fraction I); then, the free phytosterols were eluted with 2 × 5 mL of *n*-hexane/ethyl acetate (5:95, *v*/*v*) (fraction II); finally, using a gentle stream of nitrogen at 35 °C for drying fractions of I and II.

#### 2.3.5. Derivatization of Free/Combined Phytosterols


(1)Derivatization of combined phytosterols


Derivatization was carried out referring to the method of Sodeif Azadmard-Damirchi et al. [45]. Firstly, add 3 mL of KOH/CH_3_CH_2_OH solution (2 mol/L) to fraction I, mixed by vortex, saponify in a constant temperature water bath at 90 °C for 20 min, take out and cool to room temperature, add 2 mL water and 1.5 mL *n*-hexane, 2500 rpm vortex for 3 min, collect the organic phase, repeat the extraction three times, combine the organic phases, dried at 35 °C by using a gentle stream of nitrogen, add 100 μL *N*,*O*-bis(trimethylsilyl)trifluoroacetamide (containing 1% trimethylchlorosilane) derivatization reagent, mixed by vortex, derivatized in an oven at 105 °C for 15 min, cooled to room temperature, reconstituted by adding 100 μL *n*-hexane, mixed by vortex, and stored for GC analysis.


(2)Derivatization of free phytosterols


Add 100 μL *N*,*O*-bis(trimethylsilyl)trifluoroacetamide (containing 1% trimethylchlorosilane) derivatization reagent to fraction II, mixed by vortex, and derivatized in an oven at 105 °C for 15 min, cooled to room temperature, reconstituted by adding 100 μL *n*-hexane, mixed by vortex, and stored for GC analysis.

#### 2.3.6. GC Analysis of Free/Combined Phytosterols [46]

The derived free/combined phytosterols were separated by GC (Agilent 6890N, Agilent Company, Santa Clara, CA, USA) fitted with a DB-5HT capillary column (15 m × 0.25 mm × 0.25 μm, Agilent Technologies Company, Santa Clara, CA, USA). Injector: injection volume of 1 μL; injection port: heater 320 °C, pressure 9.9 psi, split ratio 25:1; carrier gas helium. Column oven: the oven temperature was initially set at 60 °C for 1.0 min and then increased at a rate of 10 °C/min to 260 °C for 14 min. FID detector: heater set at 320 °C, hydrogen flow set at a rate of 40.0 mL/min, air flow set at a rate of 400.0 mL/min, He flow set at a rate of 30.0 mL/min.

#### 2.3.7. Quantitative Analysis of Free/Combined Phytosterols

The internal standard was used for quantitative analysis, and the content of components was calculated according to the ratio of the chromatographic peak area of the tested compound to the internal standard. Calculated as follows:$$\mathrm{Phytosterol}\;\mathrm{content}\;\left(\mathrm{mg}/\mathrm{kg}\right)=\frac{Ax\times Ms\times 1000}{As\times M}$$

*Ax* is the chromatographic peak area of the tested component; *Ms* is the mass (mg) of the internal standard (β-cholestanol); *As* is the chromatographic peak area of the internal standard (β-cholestanol); *M* is the weight of the sample (g).

#### 2.3.8. Spike Recovery


(1)Spike recovery of phytosterols in rapeseed


Accurately weigh several rapeseed samples (50 mg each) and put them into glass test tubes, respectively, one of which is used as a blank control and the rest of the samples are spiked with standard mixture of brassicasterol, campesterol, β-sitosterol, and cholesterol oleate according to three concentration levels (low, medium and high). Add 50 μL of β-cholestanol (1 mg/mL in *n*-hexane) standard solution to each test tube, and use the Folch method in Section 2.3.2 to extract phytosterols. The extracted free/combined phytosterols were then separated by SPE. The SPE separation operation is performed as described in Section 2.3.3.


(2)Spike recovery of phytosterols in rapeseed oil


Accurately weigh several rapeseed oil samples (50 mg each) and put them into glass test tubes, respectively, one of which is used as a blank control and the rest of the samples are spiked with standard mixture of brassicasterol, campesterol, β-sitosterol, and cholesterol oleate according to three concentration levels (low, medium and high). Add 36 μL of β-cholestanol (1 mg/mL in *n*-hexane) standard solution to each test tube, and add 5 mL *n*-hexane to dilute the oil sample, which was then separated by SPE. The SPE separation operation is performed as described in Section 2.3.3.

The recovery is calculated as follows:$$\mathrm{Spike}\;\mathrm{recovery}\;=\frac{\left(Ms-Mx\right)}{Ma}\times 100\%$$

Ms is the measured content of the analytes in the spiked sample; Mx is the measured content of the analytes in the control sample; Ma is the content of the added standards.

#### 2.3.9. Analysis of Oil Content in Rapeseed, Residual Oil Content in Rapeseed Cake and Oil Yield in Rapeseed Oil Processing [47,48]


(1)Sample preparation method


Add 2 mL of 5% H_2_SO_4_/CH_3_OH solution and 300 μL toluene to 15 mg rapeseed or rapeseed cake samples. Additionally, add 50 μL of methyl heptadecanoate standard solution (5 mg/mL in *n*-hexane) as internal standard, then, saponify in a constant temperature water bath at 95 °C for 1.5 h, take out and cool to room temperature, add 2 mL of 0.9% NaCl aqueous solution and 1 mL of *n*-hexane, vortexed at 1800 rpm for 5 min, centrifuged at 5000 rpm for 5 min, and collect the organic phase for GC analysis.


(2)GC analysis of fatty acids


The analysis of oil content in rapeseed and residual oil content in rapeseed cake were performed by GC (Agilent 6890N, Agilent Company, Santa Clara, CA, USA) fitted with a DB-Fast FAME capillary column (30 m × 0.25 mm × 0.25 µm, Agilent Technologies Company, Santa Clara, CA, USA). Injector: injection volume of 1 μL; injection port: heater 250 °C, pressure 9.9 psi, split ratio 20:1; carrier gas helium. Column oven: the oven temperature was initially set at 80 °C for 0.5 min, and then increased at a rate of 40 °C/min to 165 °C, and then increased at a rate of 4 °C/min to 230 °C for 6 min. FID detector: heater set at 260 °C, hydrogen flow set at a rate of 40.0 mL/min, air flow set at a rate of 350.0 mL/min, He flow set at a rate of 20.0 mL/min.


(3)Oil content in rapeseed or residual oil content in rapeseed cake is calculated as follows:
$$\mathrm{Oil}\;\mathrm{content}\;\mathrm{in}\;\mathrm{rapeseed}\;\mathrm{or}\;\mathrm{residual}\;\mathrm{oil}\;\mathrm{content}\;\mathrm{in}\;\mathrm{rapeseed}\;\mathrm{cake}\;\left(\%\right)=\frac{\left(\frac{S1}{S2}\right)\times N}{M}\times 100\%$$


S1 is the total peak areas of fatty acids of the sample; S2 is the peak area of internal standard; N is the quality of internal standard; M is the quality of sample.


(4)Oil yield in rapeseed oil processing is calculated as follows:
$$\mathrm{Oil}\;\mathrm{yield}\;\left(\%\right)=\frac{\left(Ro-Co\right)}{1-Co}\times 100\%$$


Ro is the oil content of rapeseed; Co is the residual oil content of rapeseed cake.

#### 2.3.10. Data Statistics and Analysis

Excel (2010 version, Microsoft Company, Redmond, WA, USA) was used for data processing. Origin (2019 version, OriginLab Company, Northampton, MA, USA) and SPSS-20 were used for graphing and significance analysis. SPSS 20.0 (SPSS Inc., Chicago, IL, USA) was used for ANOVA and Duncan’s multiple range test (*p* < 0.05). a–d Mean values (a > b > c > d, corresponding to the same parameter) not followed by a common letter differ significantly (*p* < 0.05). Three parallel determinations in each group were represented by mean value ± standard deviation.

## 3. Results and Discussion

### 3.1. Qualitative Analysis of Phytosterols by GC

The qualitative analysis of phytosterols by GC was based on retention time of the phytosterol standards. The GC chromatograms of phytosterol standards, and free/combined phytosterols in rapeseed oil after separation by SPE were shown in Figure 3. The results showed that complete separation of the main phytosterols of rapeseeds and internal standard (β-cholestanol) was achieved with optimal GC analysis conditions, and the whole separation can be completed within 28 min; moreover, free/combined phytosterols in rapeseed oil samples can be effectively separated by SPE pretreatment. After SPE separation and enrichment, no impurity interference peaks were observed for analysis of both free and combined phytosterols in rapeseed oil samples by GC. Therefore, the analysis process is as follows: first the qualitative analysis of phytosterols in the samples could be accurately performed by comparing with the retention time of the phytosterol standards, and then accurate quantitative analysis was performed by the internal standard method as described in Section 2.3.6.

### 3.2. Optimization of the Liquid–Liquid Extraction Method

Appropriate extraction methods are a fundamental and necessary step in analysis and have a direct impact on reported results [49]. In order to select the suitable extraction method, three liquid–liquid extraction methods (including Folch, Bligh–Dyer and MTBE) were employed to extract phytosterols from rapeseed, respectively, and the extracted contents of phytosterols were compared (Figure 4). Rapeseed Dadi 199 was selected as the test sample to compare the extraction effects of different extraction methods. There are significant differences in the contents of combined phytosterols extracted from Dadi 199 rapeseed by three extraction methods (*p* < 0.05). For free phytosterols, the Folch method and the Bligh–Dyer method have similar effects on the extraction of free phytosterols. Compared with the Folch and the Bligh–Dyer method, the MTBE method showed a significant difference for the extraction of free phytosterols. The extraction effects of different extraction methods were further evaluated based on the extracted content of phytosterols in rapeseed, and the results showed that extracted by the Folch method, both the combined and free phytosterols had the highest content, and the total phytosterol content reached 3190.2 mg/kg (Appendix A), which may be due to the better solubility of phytosterols in a larger proportion of chloroform extraction solution. Therefore, the Folch method was selected as the method for extracting free/combined phytosterols from rapeseed and rapeseed cake in subsequent experiments.

### 3.3. Recovery of Phytosterols in Rapeseed and Rapeseed Oil

In order to verify the accuracy of the experiment, free/combined phytosterols were spiked and recovered in the rapeseed samples and their oil samples. The experimental results are shown in Table 1; Table 2, respectively. According to Table 1, the recoveries of the analytes in rapeseed ranged from 82.7% to 104.5%, and the RSD values were all less than 10.00%. The recoveries of brassicasterol, campesterol, β-sitosterol and cholesterol oleate were 84.9–104.3%, 97.5–104.5%, 82.8–94.9%, and 82.7–92.3%, respectively. As shown in Table 2, the recoveries of the analytes in rapeseed oil ranged from 83.8% to 116.3%, and the RSD values were all less than 10.00%. The recoveries of brassicasterol, campesterol, β-sitosterol and cholesterol oleate were 99.6–116.3%, 87.9–99.2%, 83.8–98.1%, and 91.3–94.1%, respectively. The recoveries were all within the feasible range from 80% to 120%, indicating that the method has good accuracy, which met the analysis requirements [50].

### 3.4. Effects of Microwave Pretreatment on the Composition and Content of Phytosterols in Rapeseed, Rapeseed Oil and Rapeseed Cake

#### 3.4.1. Phytosterols in Rapeseed

As shown in Figure 5 and Appendix A, it can be seen that there are mainly three phytosterols (brassicasterol, campesterol and β-sitosterol) in rapeseed, which exist in both bound and free forms. In both Zhongshuang 19 and Dadi 199 rapeseed, the highest content of free and combined phytosterols is β-sitosterol, followed by campesterol, and the lowest content of free and combined phytosterols is brassosterol. In rapeseed without microwave pretreatment, the content of total combined phytosterols (1637.05 mg/kg) was higher than that of the total free phytosterols (1483.67 mg/kg) in Zhongyouza 19 rapeseed; however, the content of total combined phytosterols (1561.33 mg/kg) was slightly lower than that of the total free phytosterols (1657.27 mg/kg) in Dadi 199 rapeseed. The results showed that both combined phytosterols and free phytosterols existed in rapeseed, and their contents were basically at the same level; and the contents of different forms of phytosterols in different varieties of rapeseed were different.

We also found that compared with untreated rapeseed, the content of total phytosterols was increased slightly in both Zhongyouza 19 (from 3120.71 to 3230.23 mg/kg) and Dadi 199 rapeseed (from 3218.61 to 3272.84 mg/kg) after microwave pretreatment. For Zhongyouza 19 rapeseed, the contents of total combined phytosterols (from 1637.05 to 1685.3 mg/kg) and total free phytosterol (from 1483.67 to 1544.9 mg/kg) were slightly increased after microwave pretreatment; however, for Dadi 199 rapeseed, the content of total combined phytosterols (from 1561.33 to 1530.4 mg/kg) was decreased slightly and the content of total free phytosterols (from 1657.27 to 1742.5 mg/kg) was slightly increased after microwave pretreatment. However, the results of statistical analysis showed that the above changes did not have significant differences, compared with untreated rapeseed, only the content of free brassinosterol and free campesterol were significantly different (*p* < 0.05) in Dadi 199 rapeseed; however, in Zhongyouza 19 rapeseed, only the content of free campesterol was significantly different (*p* < 0.05) after microwave pretreatment. These observed changes may be attributed to the destruction of the microstructure and cellular structure of the rapeseed during microwave pretreatment, making the chemical constituents in the rapeseed easier to be extracted. Therefore, the phytosterol content of rapeseed was increased after microwave pretreatment [35,38].

#### 3.4.2. Phytosterols in Rapeseed Oil

As shown in Figure 6 and Appendix A, it can be seen that compared with untreated rapeseed, after microwave pretreatment, the content of total phytosterols was significantly increased (from 5769.68 to 6266.56 mg/kg, *p* < 0.05) in pressed oil of Zhongyouza 19 rapeseeds; however, the content of total phytosterols was slightly increased (from 5653.93 to 5759.83 mg/kg) in pressed oil of Dadi 199 rapeseeds. For pressed oil of Zhongyouza 19 rapeseed, the contents of total combined phytosterols (from 3086.17 to 3334.14 mg/kg, increased by 8.03%) and total free phytosterol (from 2683.52 to 2932.42 mg/kg, increased by 9.28%) were significantly increased (*p* < 0.05) after microwave pretreatment of the rapeseeds; however, for pressed oil of Dadi 199 rapeseed, the content of total combined phytosterols (from 2791.07 to 2803.48 mg/kg) was increased slightly and the content of total free phytosterols (from 2862.86 to 2956.36 mg/kg) was significantly increased (*p* < 0.05) after microwave pretreatment of the rapeseeds. Furthermore, compared with pressed oil of untreated rapeseed, the contents of different forms of phytosterols in pressed oil of Zhongyouza 19 rapeseed increased significantly (*p* < 0.05) after microwave pretreatment of the rapeseeds; however, in pressed oil of Dadi 199 rapeseed, only the content of free brassicasterol and free campesterol increased significantly (*p* < 0.05) after microwave pretreatment of the rapeseeds. These observed changes may also be attributed to the destruction of the microstructure and cellular structure of the rapeseed during microwave pretreatment, making the chemical constituents in the rapeseed easier to enter the pressed oil.

We also found that the content of total combined phytosterols was much higher than that of the total free phytosterols in pressed oil of Zhongyouza 19 rapeseed with or without microwave pretreatment; however, the content of total combined phytosterols was slightly lower than that of the total free phytosterols in pressed oil of Dadi 199 rapeseed with or without microwave pretreatment. The results showed that different varieties of rapeseeds have different processing characteristics, the varieties of rapeseed, planting conditions, etc. will affect the quality of rapeseed oil, as well as different rapeseed processing techniques are also major factors affecting the quality of rapeseed oil.

#### 3.4.3. Phytosterols in Rapeseed Cake

As shown in Figure 7 and Appendix A, it can be seen that compared with untreated rapeseed, after microwave pretreatment, the content of total phytosterols was significantly increased (from 1722.70 to 1855.13 mg/kg, *p* < 0.05) in rapeseed cake of Dadi 199; however, the content of total phytosterols was slightly decreased (from 1405.69 to 1393.05 mg/kg) in rapeseed cake of Zhongyouza 19. For rapeseed cake of Zhongyouza 19, the contents of total combined phytosterols (from 672.64 to 688.19 mg/kg) were significantly increased (*p* < 0.05) after microwave pretreatment of the rapeseeds; however, the content of total free phytosterols was slightly decreased (from 733.06 to 704.86 mg/kg). For rapeseed cake of Dadi 199, both the contents of total combined phytosterols (from 847.07 to 954.02 mg/kg) and the content of total free phytosterols (from 875.63 to 901.10 mg/kg) were significantly increased (*p* < 0.05) after microwave pretreatment of the rapeseeds. Furthermore, the contents of different forms of phytosterols in rapeseed cake of Dadi 199 increased significantly (*p* < 0.05) after microwave pretreatment of the rapeseeds, except for free brassicasterol and free campesterol, which only increased slightly; however, in rapeseed cake of Zhongyouza 19, the content of free phytosterols (brassicasterol, campesterol and β-sitosterol) decreased significantly (*p* < 0.05), and the contents of combined phytosterols (brassicasterol, campesterol and β-sitosterol) increased slightly after microwave pretreatment of the rapeseeds.

We also found that the content of total free phytosterols was higher than that of the total combined phytosterols in rapeseed cake of Zhongyouza 19 with or without microwave pretreatment. At the same time, the content of total free phytosterols (875.6 mg/kg) was also higher than that of the total combined phytosterols (847.1 mg/kg) in rapeseed cake of Dadi 199 without microwave pretreatment; but the content of total free phytosterols (901.1 mg/kg) was slightly lower than that of the total combined phytosterols (954.1 mg/kg) in rapeseed cake of Dadi 199 after microwave pretreatment. The results showed that different varieties of rapeseeds have different processing characteristics, as the varieties of rapeseed, planting conditions, etc., will affect the quality of rapeseed cake, and different rapeseed processing techniques are also major factors affecting the quality of rapeseed cake.

### 3.5. Effect of Microwave Pretreatment on Oil Content in Rapeseed, Residual Oil Content in Rapeseed Cake and Oil Yield in Rapeseed Oil Processing

As shown in Figure 8 and Appendix A, it can be seen that microwave pretreatment of rapeseed could improve the oil content (improved 3.21% for Zhongyouza 19; and 5.38% for Dadi 199) and oil yield of rapeseed (improved 5.89% for Zhongyouza 19; and 9.68% for Dadi 199). Research results have proved that microwave pretreatment could destruct the microstructure and cellular structure of the rapeseed, making the lipids in cells more easily extracted, thereby improving the oil yield of rapeseed in the oil processing [35,38]. As for the residual oil content in rapeseed cake, it can be seen that after microwave pretreatment of Zhongyouza 19 rapeseed, the residual oil content of rapeseed cake slightly decreased (from 13.46% to 13.00%); however, the residual oil content in rapeseed cake increased slightly (from 20.33% to 20.53%) after microwave pretreatment of Dadi 199 rapeseed, but the differences are not significant.

### 3.6. Effect of Microwave Pretreatment on Migration of Phytosterols during Oil Processing

As shown in Figure 9, it can be seen that without microwave pretreatment, the free and combined phytosterols in Dadi 199 rapeseed were 51.5% and 48.5%, respectively; after microwave pretreatment, the free and combined phytosterols were 53.2% and 46.8%, respectively. The experimental results showed that the content of free phytosterols in rapeseed of Dadi 199 is higher than that of combined phytosterols, and after microwave pretreatment, the proportion of free phytosterols in rapeseed further increased, which may be due to the dissociation of a small part of the combined phytosterols into free phytosterols by microwave heat treatment.

During the oil processing of Dadi 199 rapeseed without microwave pretreatment, 28.8% and 22.7% of free phytosterols in rapeseed migrated to rapeseed oil and rapeseed cake, respectively; and 27.2% and 21.3% of combined phytosterols in rapeseed migrated to rapeseed oil and rapeseed cake, respectively. During the oil processing of Dadi 199 rapeseed with microwave pretreatment, 31.4% and 21.8% of free phytosterols in rapeseed migrated to rapeseed oil and rapeseed cake, respectively; and 26.4% and 20.4% of combined phytosterols in rapeseed migrated to rapeseed oil and rapeseed cake, respectively. The experimental results showed that during the oil processing of Dadi 199 rapeseed without microwave pretreatment, 56.0% and 44.0% of total phytosterols (combined and free) migrated into rapeseed oil and rapeseed cake, respectively; and after microwave pretreatment of rapeseed, the proportions were 57.8% and 42.2%, respectively. Therefore, it can be concluded that more than 55% of the total phytosterols in Dadi 199 rapeseed (with or without microwave) were transferred into the rapeseed oil; and after microwave pretreatment of rapeseed, the proportion of phytosterols transferred into rapeseed oil increased slightly. Furthermore, we also found that during oil processing, both free and combined phytosterols migrate more easily into the rapeseed oil instead of rapeseed cake, which may be related to the solubility of phytosterols in oil.

As shown in Figure 10, it can be seen that without microwave pretreatment, the free and combined phytosterols in Zhongyouza 19 rapeseed were 47.5% and 52.5%, respectively; after microwave pretreatment, the free and combined phytosterols were 47.8% and 52.2%, respectively. The results showed that the content of free phytosterols in rapeseed of Zhongyouza 19 is lower than that of combined phytosterols, and after microwave pretreatment, the proportion of free phytosterols in rapeseed slightly increased, which may be due to the dissociation of a small part of the combined phytosterols into free phytosterols by microwave heat treatment.

During the oil processing of Zhongyouza 19 rapeseed without microwave pretreatment, 29.8% and 17.7% of free phytosterols in rapeseed migrated to rapeseed oil and rapeseed cake, respectively; and 35.6% and 16.9% of combined phytosterols in rapeseed migrated to rapeseed oil and rapeseed cake, respectively. During the oil processing of Zhongyouza 19 rapeseed with microwave pretreatment, 32.3% and 15.5% of free phytosterols in rapeseed migrated to rapeseed oil and rapeseed cake, respectively; and 37.0% and 15.2% of combined phytosterols in rapeseed migrated to rapeseed oil and rapeseed cake, respectively. The experimental results showed that during the oil processing of Zhongyouza 19 rapeseed without microwave pretreatment, 65.4% and 34.6% of total phytosterols (combined and free) migrated into rapeseed oil and rapeseed cake, respectively; and after microwave pretreatment of rapeseed, the proportions were 69.3% and 30.7%, respectively. Therefore, it can be concluded that more than 65% of the total phytosterols in Zhongyouza 19 rapeseed (with or without microwave) were transferred into the rapeseed oil; and after microwave pretreatment of rapeseed, the proportion of phytosterols transferred into rapeseed oil increased significantly. Furthermore, we also found that during oil processing, both free and combined phytosterols migrate more easily into the rapeseed oil instead of rapeseed cake.

In summary, whether it is the rapeseed of Zhongyouza 19 or Dadi 199, during the oil processing, the total phytosterols (combined and free) in rapeseed migrated to rapeseed oil and rapeseed cake, respectively, in which more than 55% of the total phytosterols migrated into rapeseed oil. The experimental results showed that after microwave pretreatment of the rapeseeds, not only the oil yield of rapeseed, but also the content of phytosterols in rapeseed oil could be increased [35,38,45]. Furthermore, the proportions of combined and free phytosterols in the rapeseeds of Zhongyouza 19 and Dadi 199 were different, and the proportions of phytosterols migrated into rapeseed cake and rapeseed oil during processing were also different, which may be due to different varieties of rapeseeds have different processing characteristics, the varieties of rapeseed, planting conditions, etc. will affect the quality of rapeseed oil and rapeseed cake.

## 4. Conclusions

In this study, a simple, efficient, and rapid SPE–GC method was developed for the simultaneous analysis of free/combined phytosterols in rapeseed and their dynamic changes during microwave pretreatment and oil processing. The established method was applied to study the dynamic changes of the form and content of phytosterols in rapeseed and its products (rapeseed oil and cake) during rapeseed microwave pretreatment and the oil production process. The results showed that more than 55% of the free/combined phytosterols in rapeseed were transferred to rapeseed oil during the oil processing, and this proportion will increase after microwave pretreatment of rapeseed. Furthermore, we found that the proportions of combined and free phytosterols were different in the rapeseeds of Zhongyouza 19 and Dadi 199, and the proportions of phytosterols migrated into rapeseed cake and rapeseed oil during processing were different. Thus, it could be concluded that the varieties of rapeseed and planting/processing conditions will affect the quality of rapeseed oil and rapeseed cake. Collectively, this work will provide analytical methods and data support for a comprehensive understanding of phytosterols in rapeseed and its products during oil processing.

## Figures and Tables

**Figure 1 foods-11-03219-f001:**
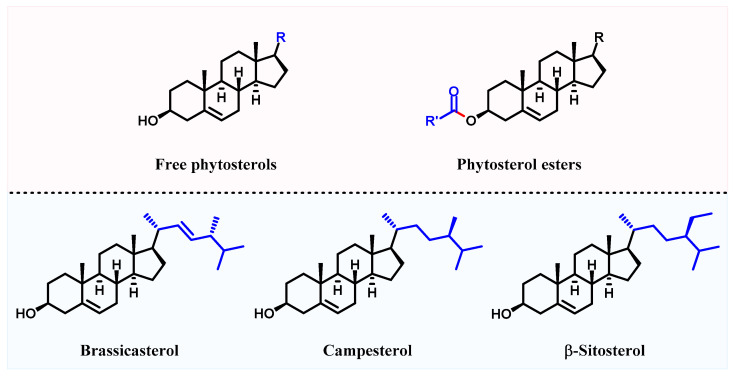
The chemical structures of free phytosterols, phytosterol esters, brassicasterol, campesterol and β-sitosterol.

**Figure 2 foods-11-03219-f002:**
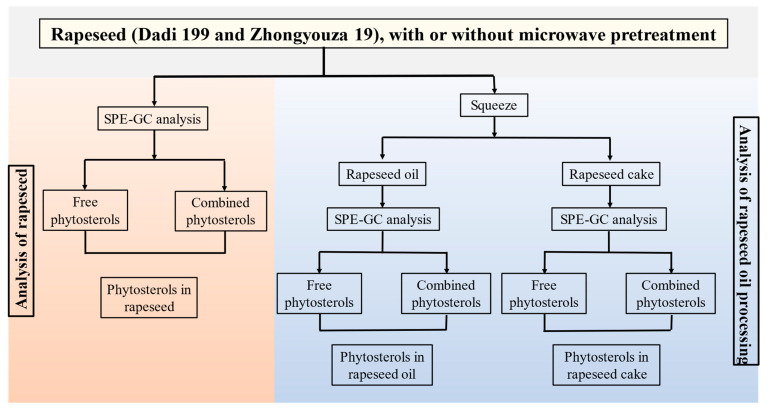
Schematic diagram for analysis the content of free and combined phytosterols in rapeseed and its products (rapeseed oil and cake) during rapeseed microwave pretreatment and the oil production process.

**Figure 3 foods-11-03219-f003:**
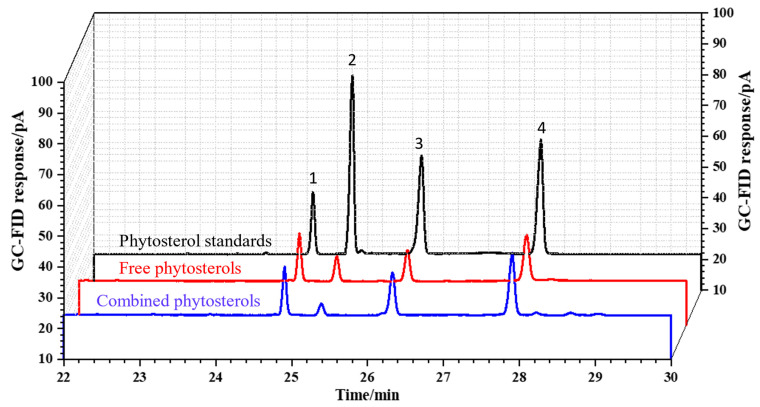
GC chromatogram for analysis of phytosterol standards (black line), and free/combined phytosterols in rapeseed oil after separation by SPE: free phytosterols in rapeseed oil (red line) and combined phytosterols (after saponification) in rapeseed oil (blue line). Peak 1, β-cholestanol; Peak 2, brassicasterol; Peak 3, campesterol; Peak 4, β-sitosterol.

**Figure 4 foods-11-03219-f004:**
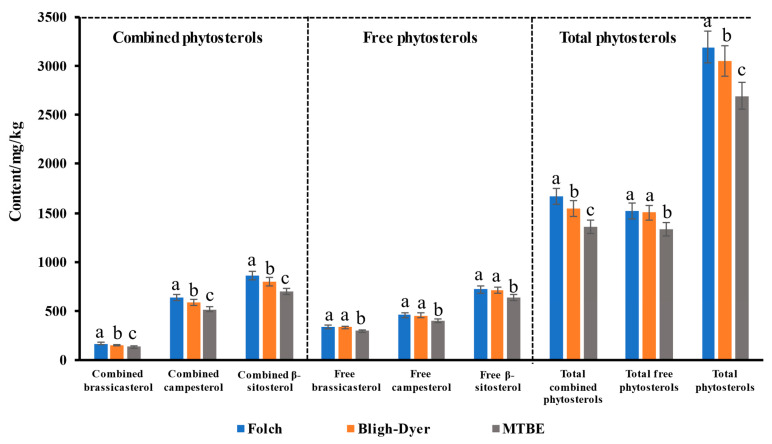
Optimization of the liquid–liquid extraction method. Values are the means ± standard deviations; *n* = 3 parallel determinations. a–c Mean values (a > b > c, corresponding to the same parameter) not followed by a common letter differ significantly (*p* < 0.05).

**Figure 5 foods-11-03219-f005:**
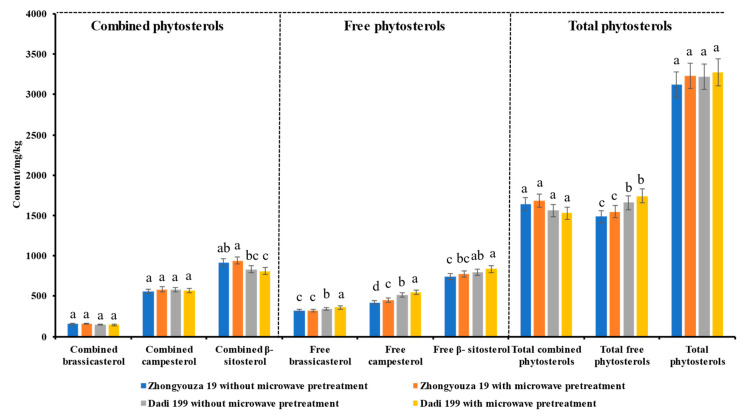
Effects of microwave pretreatment on the composition and content of phytosterols in rapeseed. Values are the means ± standard deviations; *n* = 3 parallel determinations. a–d Mean values (a > b > c > d, corresponding to the same parameter) not followed by a common letter differ significantly (*p* < 0.05).

**Figure 6 foods-11-03219-f006:**
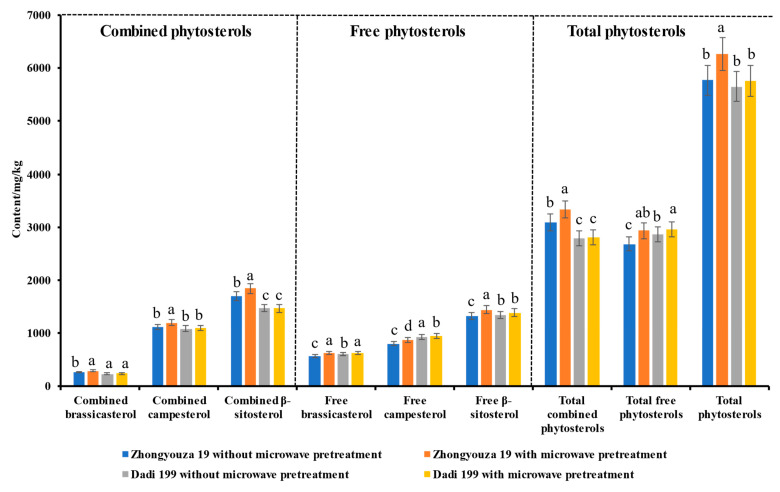
Effects of microwave pretreatment on the composition and content of phytosterols in rapeseed oil. Values are the means ± standard deviations; *n* = 3 parallel determinations. a–d Mean values (a > b > c > d, corresponding to the same parameter) not followed by a common letter differ significantly (*p* < 0.05).

**Figure 7 foods-11-03219-f007:**
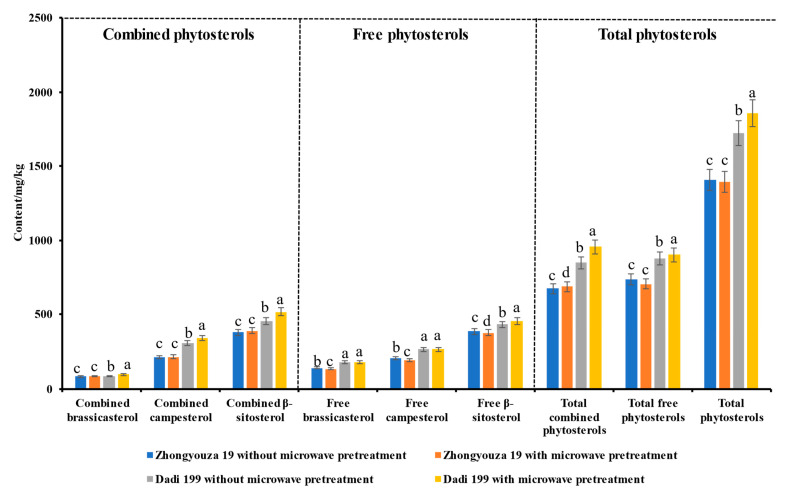
Effects of microwave pretreatment on the composition and content of phytosterols in rapeseed cake. Values are the means ± standard deviations; *n* = 3 parallel determinations. a–d Mean values (a > b > c > d, corresponding to the same parameter) not followed by a common letter differ significantly (*p* < 0.05).

**Figure 8 foods-11-03219-f008:**
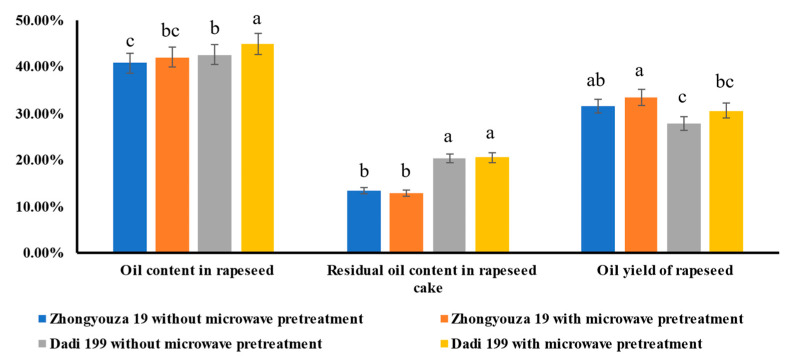
Analysis of the oil content in rapeseed, the residual oil content in rapeseed cake and the oil yield in oil processing with/without microwave pretreatment of rapeseed. Values are the means ± standard deviations; *n* = 3 parallel determinations. a–c Mean values (a > b > c, corresponding to the same parameter) not followed by a common letter differ significantly (*p* < 0.05).

**Figure 9 foods-11-03219-f009:**
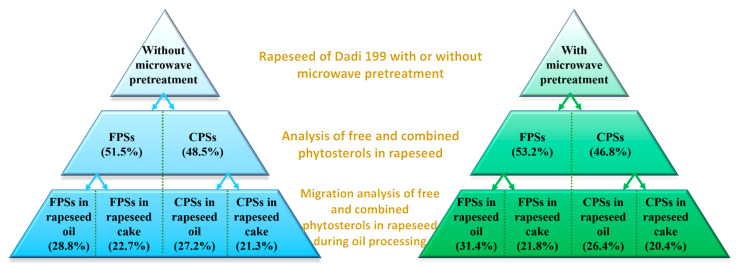
The dynamic changes of combined and free phytosterols in rapeseed of Dadi 199 during oil processing with/without microwave pretreatment of rapeseed. CPSs: combined phytosterols; FPSs: free phytosterols. Values are the means ± standard deviations; n = 3 parallel determinations.

**Figure 10 foods-11-03219-f010:**
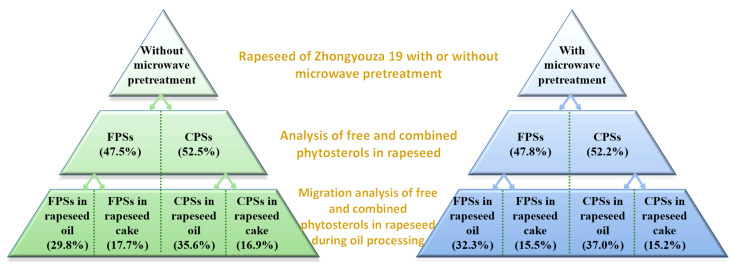
The dynamic changes of combined and free phytosterols in rapeseed of Zhongyouza 19 during oil processing with/without microwave pretreatment of rapeseed. CPSs: combined phytosterols; FPSs: free phytosterols. Values are the means ± standard deviations; *n* = 3 parallel determinations.

**Table 1 foods-11-03219-t001:** Recovery of phytosterols in rapeseed.

Compounds	Original Content in Rapeseed (μg/g)	Add Content (μg/g)	Final Content (μg/g)	Recovery (%)	Average Recovery (%)	RSD (%)
Brassicasterol	319.30	134	433.09	84.9	96.9	3.70
		660	988.22	101.4		3.94
		1320	1696.02	104.3		0.56
Campestrol	421.45	100	525.93	104.5	100.4	5.94
		500	917.48	99.2		6.78
		1000	1396.54	97.5		2.06
β-Sitosterol	742.92	100	833.74	90.8	89.5	9.74
		500	1217.55	94.9		3.95
		1000	1570.46	82.8		2.06
Cholesteryl oleate	0	2.97	2.74	92.3	87.2	0.72
		14.85	12.84	86.5		4.94
		29.69	24.56	82.7		2.19

Values are the means ± standard deviations; *n* = 3 parallel determinations.

**Table 2 foods-11-03219-t002:** Recovery of phytosterols in rapeseed oil.

Compounds	Original Content in Rapeseed Oil (μg/g)	Add Content (μg/g)	Final Content (μg/g)	Recovery (%)	Average Recovery (%)	RSD (%)
Brassicasterol	364.18	240	604.55	100.2	105.3	3.47
		480	922.60	116.3		4.18
		960	1320.05	99.6		4.16
Campestrol	689.91	380	1023.73	87.9	93.8	2.91
		760	1444.13	99.2		1.20
		1520	2123.81	94.3		3.50
β-Sitosterol	1213.55	860	1934.35	83.8	90.9	8.96
		1720	2900.21	98.1		4.40
		3440	4337.15	90.8		1.79
Cholesteryl oleate	0	10.69	9.86	92.3	92.6	1.29
		21.39	19.53	91.3		0.60
		42.76	40.23	94.1		4.70

Values are the means ± standard deviations; *n* = 3 parallel determinations.

## Data Availability

The data presented in this article are available on reasonable request, from the corresponding author.

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
