# Peer review of "Simultaneous Analysis of Free/Combined Phytosterols in Rapeseed and Their Dynamic Changes during Microwave Pretreatment and Oil Processing"

_foods, 2022, doi:10.3390/foods11203219_

Round 1

Reviewer 1 Report

·       A key revision is explaining why the quantification approach was used.  Why an IS (a structural analogue) was used to quantify.  Why single point calibration was used for quantification using IS rather than conventional multi-point standard curve, using authentic standards while the IS is used in conjunction with analyte standards. It is very reasonable to do that even if there is no analyte free matrix.  The literature is rich on quantifying sterols using validated method with multi-point calibration being used.  So, a clear justification for the approach is needed.  Also, the authors must show that single point calibration results in quantitative values within known linear range [what is it?]. If the authors can justify their approach, then the paper has a merit. 

·       The work is not absolute quantification, it is a rough estimation of the true value (semi-quantification).  Only using fully validated methods will give as true value as possible. The reported quantification approach is fit-for-purpose and that needs explanation why this approach fit the purpose of the study.  Why rough estimation of the value is sufficient to inform the research question.

·        The FDA health claim for phytosterols does not extend to brassicasterol.  This one is not included in the FDA claim.  It does not mean that it possibly have similar health effects but there are no clinical trails that support this claim. The paper needs to state that clearly.

·       Sample pre-treatment needs expansion and clarifications – it is not sufficient to cite past papers.

·       Did the authors consider that microwave heating will generate harmful phytosterol oxidation products.  This is a proven fact and it needs to be highlighted. In fact, without measuring the oxidation products, we probably need to shy away from microwaving the sample as part of the extraction process. We will be harming the people using such extract.

·       Is it possible that microwaving has broken the ester bond and resulted in more free sterols.

·       Pg 7, it was not clear how fatty acids were measured and why.

·       Since the method is semi-quantitative (as single point calibration with IS), it is impossible to state values were reduced or increased when they are very close to each other.  This needs to be articulated.

Author Response

Responses to comments from reviewer #2:

Reviewer: 2

Thank you very much for your comments and suggestions.

Comments:

  1. A key revision is explaining why the quantification approach was used. Why an IS (a structural analogue) was used to quantify.  Why single point calibration was used for quantification using IS rather than conventional multi-point standard curve, using authentic standards while the IS is used in conjunction with analyte standards. It is very reasonable to do that even if there is no analyte free matrix.  The literature is rich on quantifying sterols using validated method with multi-point calibration being used.  So, a clear justification for the approach is needed.  Also, the authors must show that single point calibration results in quantitative values within known linear range [what is it?]. If the authors can justify their approach, then the paper has a merit.
    Response: Thank you very much for this comment.

Response: Thanks for your comments. The change of injection volume and the slight change of chromatographic conditions had little effect on the quantitative results of internal standard method. Especially, adding internal standard before sample pretreatment (such as concentration, extraction, derivatization, etc.) and then pretreatment can partially compensate the loss of the components to be measured during sample pretreatment. At the same time, GC-FID has almost no matrix effect, and the results of single point calibration are consistent with conventional multi-point standard curve. In lipid analysis, internal standard method is often used to quantify lipids, which is one of the common quantitative methods. The choice of internal standard is usually to select structural analogues to quantify a certain type of lipid (β-cholestanol is used as internal standard in this paper). Compared with the external standard method, the internal standard method can better quantitatively analyze each sample and avoid the error caused by the instability of the machine. For example, Richard Broughton et al [1] determined the content of total sterols in seed oil by GC-FID with cholesterol as internal standard. Using 5β-cholestan-3α-ol as internal standard, Samantha Duong et al [2] determined total sterols in fortified foods by GC-FID.

[1] Broughton R; Ruiz-Lopez N; Hassall KL; Martinez-Force E; Garces R; Salas JJ; Beaudoin F. New insights in the composition of wax and sterol esters in common and mutant sunflower oils revealed by ESI-MS/MS. Food Chem. 2018, 269, 70-79.

[2] Duong S; Strobel N; Buddhadasa S; Stockham K; Auldist M; Wales B; Orbell J; Cran M. Rapid measurement of phytosterols in fortified food using gas chromatography with flame ionization detection. Food Chem. 2016, 211, 570-576.

  1. The work is not absolute quantification, it is a rough estimation of the true value (semi-quantification). Only using fully validated methods will give as true value as possible. The reported quantification approach is fit-for-purpose and that needs explanation why this approach fit the purpose of the study.  Why rough estimation of the value is sufficient to inform the research question.

Response: Thanks for your comments. In lipid analysis, due to the large number of lipid molecules, a single corresponding internal standard cannot be used for absolute quantification. Semi-quantitative method is the most common method to study lipid quantification. For example, PC-d7 is used to quantify all pc molecules. In past articles in our laboratory, there have been semi-quantitative applications of such methods for fatty acids and all intact lipid molecules [1-2].

[1] Wu B; Xie Y; Xu S; Lv X; Yin H; Xiang J; Chen H; Wei F. Comprehensive Lipidomics Analysis Reveals the Effects of Different Omega-3 Polyunsaturated Fatty Acid-Rich Diets on Egg Yolk Lipids. J. Agric. Food Chem. 2020, 68, 15048-15060.

[2] Xu SL; Wu BF; Oresic M; Xie Y; Yao P; Wu ZY; Lv X; Chen H; Wei F. Double Derivatization Strategy for High-Sensitivity and High-Coverage Localization of Double Bonds in Free Fatty Acids by Mass Spectrometry. Anal. Chem. 2020, 92, 6446-6455.

  1. The FDA health claim for phytosterols does not extend to brassicasterol. This one is not included in the FDA claim. It does not mean that it possibly have similar health effects but there are no clinical trails that support this claim. The paper needs to state that clearly.

Response: Thanks for your suggestion. we have cleared the “phytosterols (excluding brassicasterol) have a positive effect on the prevention of cardiovascular disease and diabetes, and the FDA (Food and Drug Administration) has approved a health claim for phytosterols as cholesterol-lowering agents” in the revised manuscript. In addition, phytosterols have been recognized by the Food and Drug Administration (FDA) as “lowering blood cholesterols and preventing arteriosclerosis” from “Food labeling: health claims; plant sterol/stanol esters and coronary heart disease. Food and Drug Administration, HHS. Interim final rule. Federal register, 2000, 65, 54686.”

  1. Sample pre-treatment needs expansion and clarifications – it is not sufficient to cite past papers.

Response: Thanks for your comments. We have clarified the sample pretreatment process in the revised manuscript.

(1) Two varieties of rapeseed, Zhongyouza 19 and Dadi 199, were pretreated with microwave, referring to the method of Q. Zhou [39] and others [34, 38] with slight modifications. After cleaning and impurity removal, the rapeseed was adjusted to 10% moisture, placed in refrigerator at 4 ℃ for 12 h, weighed 400 g, divided into 8 plates with a diameter of 9 cm, and placed in a microwave oven at a microwave power of 800 W for 7 min, quickly cooled to room temperature, and rapeseed without microwave treatment was used as a blank control.

(2) Drying of rapeseed. An appropriate amount of rapeseed was pre-frozen in an ultralow temperature refrigerator for 6 h, and then freeze-dried by a vacuum freeze dryer to obtain a dry base of rapeseed, which was stored in refrigerator at -20 ℃ until further process.

(3) Preparation of cold-pressed canola oil (two varieties of rapeseeds, Zhongyouza 19 and Dadi 199). Dried non-microwave rapeseed and microwaved rapeseed were placed in refrigerator at 4 ℃ for 12 h, and the moisture content was adjusted to 6 %. Take 100 g dried non-microwave rapeseed and microwaved rapeseed and press with LTP-205 Liangtai oil press to obtain rapeseed oil and rapeseed cake, which are stored in refrigerator at 4 °C until further process.

  1. Did the authors consider that microwave heating will generate harmful phytosterol oxidation products. This is a proven fact and it needs to be highlighted. In fact, without measuring the oxidation products, we probably need to shy away from microwaving the sample as part of the extraction process. We will be harming the people using such extract.

Response: Thanks for your comments. Research shows that up to 6 min of microwaving, no phytosterol oxidation products were detected in phytosterol mixture and a liquid mixture of phytosterols and triolein (PS + triacylglycerol, 1:100, w/w) [1]. By referring to many references in this paper, the pretreatment of rapeseed using a microwave power of 800W for 7 minutes. Yanxia Cong [2] studied the microwave pretreatment of sinapic acid derivatives in rapeseed and its role in improving the oxidative stability of rapeseed oil. Additionally, the tocopherols, sterols and canolol contents, along with the induction period of microwaved rapeseed oil increased by 3.79%, 10.0%, 76.8 times and 38.7%, respectively. Microwave pretreatment has been considered to improve the quality of rapeseed oil for canolol generation. In terms of enterprise application, Wuhan Zhongyou Kangni Technology Co. LTD is a scientific and technological service enterprise led by the Institute of Oil Crops, Chinese Academy of Agricultural Sciences. Its 7D functional rapeseed oil also uses microwave multi-effect conditioning (Double heat treatment with microwave and hot wind) technology to slightly expand and destroy the cell structure, improve the oil yield of low temperature pressing and promote the outflow of nutrients.

[1] Leal-Castaneda EJ; Inchingolo R; Cardenia V; Hernandez-Becerra JA; Romani S; Rodriguez-Estrada MT; Galindo HS. Effect of Microwave Heating on Phytosterol Oxidation. J. Agric. Food Chem. 2015, 63, 5539-5547.

[2] Cong Y; Zheng M; Huang F; Liu C; Zheng C. Sinapic acid derivatives in microwave-pretreated rapeseeds and minor components in oils. J. Food Compos. Anal. 2020, 87, 103394-103402.

  1. Is it possible that microwaving has broken the ester bond and resulted in more free sterols.

Response: Thanks for your comments. As shown in Fig. 5 in manuscript. For Zhongyouza 19 and Dadi 199 rapeseed, the results of statistical analysis showed that the total combined phytosterols and total free phytosterol did not have significant differences of the rapeseed during microwave pretreatment. In addition, the content of each combined phytosterol in rapeseed was no significant difference, no matter whether it was pretreated by microwave or not. As a result, the destruction of the microstructure and cellular structure of the rapeseed during microwave pretreatment, making the chemical constituents in the rapeseed easier to be extracted. However, the ester group of combined phytosterols (phytosterol esters) were not destroyed of the rapeseed during microwave pretreatment.

  1. Pg 7, it was not clear how fatty acids were measured and why.

Response: Thanks for your comments. We have described the process of fatty acid determination in 2.3.8. As shown in References 47 and 48, because most components of oil in a sample are composed of fatty acids and their derivatives, and the derivatives of fatty acids will also get fatty acids after hydrolysis, so the level of fatty acids can reflect the oil content of the sample.

  1. Since the method is semi-quantitative (as single point calibration with IS), it is impossible to state values were reduced or increased when they are very close to each other. This needs to be articulated.

Response: Thanks for your comments. Actually, GC-FID has almost no matrix effect, and the results of single point calibration are consistent with conventional multi-point standard curve. In lipid analysis, internal standard method is often used to quantify lipids, which is one of the common quantitative methods. The choice of internal standard is usually to select structural analogues to quantify a certain type of lipid (β-cholestanol is used as internal standard in this paper). Compounds with similar structures tend to have the approximate response in chromatography, so semi-quantification is now the main quantitative method for lipid analysis. As a commonly used quantitative method, semi-quantification is quantified for each individual sample, so it can identify the increase and decrease of content between different samples.

Reviewer 2 Report

A few recommendations for the authors are given attached

Author Response

Responses to comments from reviewer #1:

Reviewer: 1

Thank you very much for your positive comments and suggestions.

Comments:

  1. Line 40/the word “are” is missing…

“…The difference is that phytosterols are unable to be synthesized endogenously….”
Response: Thanks for your suggestion. Sorry for our mistake, line 40, we have modified the “phytosterols unable to” to “phytosterols are unable to” in the revised manuscript (Line 41).

Lines 49: Please explain FDA abbreviation in bracke (perhaps to add a list of abbreviations at the end…)

Response: Thanks for your suggestion. we have modified the “FDA” to “FDA (Food and Drug Administration)” in the revised manuscript (Line 41).

  1. Please include References for 2.3.5 (GC analysis of free/combined phytosterols) and 2.3.6 (GC analysis of fatty acids)

Response: Thanks for your suggestion. We have added reference of 46 for 2.3.5 (GC analysis of free/combined phytosterols) and references of 47-48 for 2.3.8 (GC analysis of fatty acids), respectively.

[46] Yu G; Guo T; Huang Q. Preparation of rapeseed oil with superhigh canolol content and superior quality characteristics by steam explosion pretreatment technology. Food Sci. Nutr. 2020, 8, 2271-2278.

[47] Cong Y; Zheng M; Huang F; Liu C; Zheng C. Sinapic acid derivatives in microwave-pretreated rapeseeds and minor components in oils. J. Food Compos. Anal. 2020, 87, 103394-103402.

[48] Li Y; Beisson F; Pollard M; Ohlrogge J. Oil content of Arabidopsis seeds: the influence of seed anatomy, light and plant-to-plant variation. Phytochemistry 2006, 67, 904-915.

  1. Please include some more details in 2.3.9 (3.9. Data statistics and analysis)/number of replicates?/ANOVA test used?/order of activity? e.g. a>b>c…>).

Response: Thank you very much for this suggestion. We have added more details in 2.3.9. SPSS 20.0 (SPSS Inc., Chicago, IL, USA) was used for ANOVA and Duncan's multiple range test (p<0.05). a-d Mean values (a>b>c>d, corresponding to the same parameter) not followed by a common letter differ significantly (P<0.05). Three parallel experiments in each group were represented by mean value ± standard deviation.

  1. Figures are clear and well-designed but the legends should also indicate how results presented…For instance mean value +- SD, n=?

Response: Thank you very much for this suggestion. Legends of all figures have been revised according to your suggestion in the revised manuscript. Values are means ± standard deviations, n=3 parallel determinations. a-d Mean values (a>b>c>d, corresponding to the same parameter) not followed by a common letter differ significantly (P<0.05).

  1. General comment: The session gives a clear description of the results but no reference to any work previously done in the area so that reader may understand whether findings are expected or not. For instance, lines 492-494 note: “The experimental results showed that after microwave pretreatment of the rapeseeds, not only the oil yield of rapeseed, but also the content of phytosterols in rapeseed oil could be increased.” Can you please elaborate in link to literature in the field? Please check the whole section and quote appropriate references when needed to compare to findings of current work.

Response: Thanks for your comments. As shown in reference of 35, 38 and 45 in the revised manuscript. Effect of microwave pretreatment of rapeseed on oil extraction yield by pressing, nutraceuticals content and oil stability were investigated and compared with untreated rapeseed as a control sample. Results showed that microwave pretreatment can increase oil extraction yield.

[35] Wroniak M; Rękas A; Siger A; Janowicz M. Microwave pretreatment effects on the changes in seeds microstructure, chemical composition and oxidative stability of rapeseed oil. LWT-Food Sci. Technol. 2016, 68, 634-641.

[38] Niu Y; Rogiewicz A; Wan C; Guo M; Huang F; Slominski BA. Effect of microwave treatment on the efficacy of expeller pressing of Brassica napus rapeseed and Brassica juncea mustard seeds. J. Agric. Food Chem. 2015, 63, 3078-3084.

[45] Azadmard-Damirchi S; Habibi-Nodeh F; Hesari J; Nemati M; Achachlouei BF. Effect of pretreatment with microwaves on oxidative stability and nutraceuticals content of oil from rapeseed. Food Chem. 2010, 121, 1211-1215.

  1. Line 507/Please reformulate…normally we do not start a sentence with “and”!

Response: Thank you, we have revised this sentence as suggested.

  1. Lines 509-510….Please reformulate…we may not use “furthermore” and “also” next to each other.

Response: Suggested correction has been made in the revised manuscript.

  1. Lines 509-514: Please reformulate…very large sentence! not easy for the reader to “digest”

Response: We have reformulated this large sentence to make it easier for readers to understand.